# Patient groups in Rheumatoid arthritis identified by deep learning respond differently to biologic or targeted synthetic DMARDs

**Maria Kalweit**[1], **Andrea M. Burden**[2], **Joschka Boedecker**[1], **Thomas Hügle**[3], **Theresa Burkard**[2]*

**1** Department of Computer Science, University of Freiburg, Freiburg, Germany, **2** ETH Zurich, Department of Chemistry and Applied Biosciences, Zurich, Switzerland, **3** Department of Rheumatology, Lausanne University Hospital, and University of Lausanne, Lausanne, Switzerland

* theresa.burkard@pharma.ethz.ch

**Data Availability Statement:** Data cannot be shared publicly because the data belongs to a third party (Swiss Clinical Quality Management in Rheumatic Diseases registry, SCQM) and are only

## Abstract

Cycling of biologic or targeted synthetic disease modifying antirheumatic drugs (b/tsDMARDs) in rheumatoid arthritis (RA) patients due to non-response is a problem preventing and delaying disease control. We aimed to assess and validate treatment response of b/tsDMARDs among clusters of RA patients identified by deep learning. We clustered RA patients clusters at first-time b/tsDMARD (cohort entry) in the Swiss Clinical Quality Management in Rheumatic Diseases registry (SCQM) [1999–2018]. We performed comparative effectiveness analyses of b/tsDMARDs (ref. adalimumab) using Cox proportional hazard regression. Within 15 months, we assessed b/tsDMARD stop due to non-response, and separately a ≥20% reduction in DAS28-esr as a response proxy. We validated results through stratified analyses according to most distinctive patient characteristics of clusters. Clusters comprised between 362 and 1481 patients (3516 unique patients). Stratified (validation) analyses confirmed comparative effectiveness results among clusters: Patients with ≥2 conventional synthetic DMARDs and prednisone at b/tsDMARD initiation, male patients, as well as patients with a lower disease burden responded better to tocilizumab than to adalimumab (hazard ratio [HR] 5.46, 95% confidence interval [CI] [1.76–16.94], and HR 8.44 [3.43–20.74], and HR 3.64 [2.04–6.49], respectively). Furthermore, seronegative women without use of prednisone at b/tsDMARD initiation as well as seropositive women with a higher disease burden and longer disease duration had a higher risk of non-response with golimumab (HR 2.36 [1.03–5.40] and HR 5.27 [2.10–13.21], respectively) than with adalimumab. Our results suggest that RA patient clusters identified by deep learning may have different responses to first-line b/tsDMARD. Thus, it may suggest optimal first-line b/tsDMARD for certain RA patients, which is a step forward towards personalizing treatment. However, further research in other cohorts is needed to verify our results.

available through a contract with them. Furthermore, access to SCQM data requires a collaboration with a certified rheumatologist practicing in Switzerland who also contributes data to SCQM. Finally, data access further requires ethical approval or a waiver of such issued by a local ethics committee. Researchers interested in SCQM data and willing to comply with aforementioned restrictions will be able to obtain access to the data in the same manner as the authors. Contact information of SCQM can be found at their web page at https://www.scqm.ch/. The SAS code written by TB and the Python code written by MK for this project is available from https://github.com/tiozab/comparative-effectivness-in-RA-patient-clusters.

**Funding:** The author(s) received no specific funding for this work.

**Competing interests:** I have read the journal's policy and the authors of this manuscript have the following competing interests: TH has received research grants and honorariums from Pfizer, AbbVie, Novartis, and Janssen, and does consultancy for Eli Lilly, Janssen, and Menarini; no other relationships or activities that could appear to have influenced the submitted work.

## Author summary

Rheumatoid arthritis (RA) is an auto-immune disease affecting the joints of the body. RA is subject to poor treatment response to advanced antirheumatic therapy. While previous studies have used machine learning techniques to identify different RA patient populations, no study has used these populations to evaluate potentially different treatment response of advanced RA treatments such as tumor necrosis factor inhibitors. Using machine learning, we identified five distinct RA patient groups which mainly differed by sex, disease burden/duration, and concomitant traditional RA treatment use (i.e. prednisone, methotrexate). Patients with high frequency of use of traditional RA treatment use at advanced RA treatment initiation, male patients, as well as patients with a lower disease burden responded better to tocilizumab than to adalimumab. Furthermore, seronegative women without use of prednisone at advanced RA treatment initiation as well as seropositive women with a higher disease burden and longer disease duration had a higher risk of non-response with golimumab than with adalimumab. The results are a step towards personalizing treatment and shall encourage other researchers to embrace machine learning techniques to improve treatment response in RA and other disease areas.

## Introduction

Rheumatoid arthritis (RA) is an heterogenic inflammatory disorder, often presenting a fluctuant disease activity over the disease course [1]. Despite the advent of validated treatment recommendations, it remains challenging in clinical practice to avoid disease flares or overmedication [2]. Moreover, specific biomarkers to guide optimal biologic or targeted synthetic disease modifying antirheumatic drug (b/tsDMARD) use are not available [3].

Intelligent clinical decision support systems (CDSS) use machine learning to generate personalized treatment strategies. For example, deep learning methods have shown great potential for disease prediction in the medical domain, [4,5] including the field of RA [6]. We have recently described an adaptive supervised deep learning method to predict the disease activity score using 28 joints (DAS28) erythrocyte sedimentation rate (esr) values at the next visit with a variation of 8% compared to the individual true values [7]. Furthermore, machine learning models taking into account data from genomic or microbiome data are increasingly used to classify the response to methotrexate or bDMARDs [8,9]. Machine learning has also been used in RA, using clinical data [10–13] or additional tissue samples for example [14]. We hypothesize that the use of machine learning helps to identify homogeneous RA patient groups within this heterogeneous disease which may help with the prediction of treatment response. However, to date, no study has assessed treatment response studies in machine learned clusters. Yet, it may prove crucial to use clustering approaches for digital stewardship for the treatment of RA.

This study aimed to identify b/tsDMARDs that work better or less well for certain patient groups, and thereby preventing treatment cycling and allowing disease control. Thus, we applied deep learning clustering techniques to identify RA patient groups among which we subsequently assessed comparative effectiveness of b/tsDMARDs.

## Methods

### Ethics statement

Ethical approval for this study was obtained from the CER-VD (La Commission cantonale d'éthique de la recherche sur l'être humain, ID 2020–00033). All patients provided written informed consent.

## Public and patient involvement

Neither patients nor the public was involved in the conduct of this study.

## Study design and data source

We conducted a cohort study using data derived from patients in the Swiss Clinical Quality Management in Rheumatic Diseases (SCQM) registry. The SCQM registry was established in 1997 and is used to prospectively follow RA patients [15]. Regulatory health authorities in Switzerland have recommended continuous monitoring with the SCQM system for all patients receiving b/tsDMARDs [16]. Further information is available in S1 Text.

## Study population

We identified all patients with an RA diagnosis who started their first b/tsDMARD between 1999 and 2018 while under observation in SCQM. We included patients who initiated a b/tsDMARD after or on their first visit in SCQM but not those who initiated a b/tsDMARD before their first visit in SCQM. The date of the first b/tsDMARD start is further called cohort entry. We excluded patients without a record of DAS28-esr score ≤6 months prior to their first b/tsDMARD start. An overview of the study conception is provided in Fig 1.

## Exposures

The exposure was defined as the first bDMARD (TNF inhibitors: adalimumab, certolizumab pegol, etanercept, infliximab, golimumab; non-TNF inhibitors: abatacept, rituximab, tocilizumab), or tsDMARD (baricitinib, tofacitinib). Comparative effectiveness analyses were performed on a class effect basis and on an individual b/tsDMARD basis. The reference group in the class effect analyses was TNF inhibitors, and the reference group in the individual b/tsDMARD analyses was adalimumab.

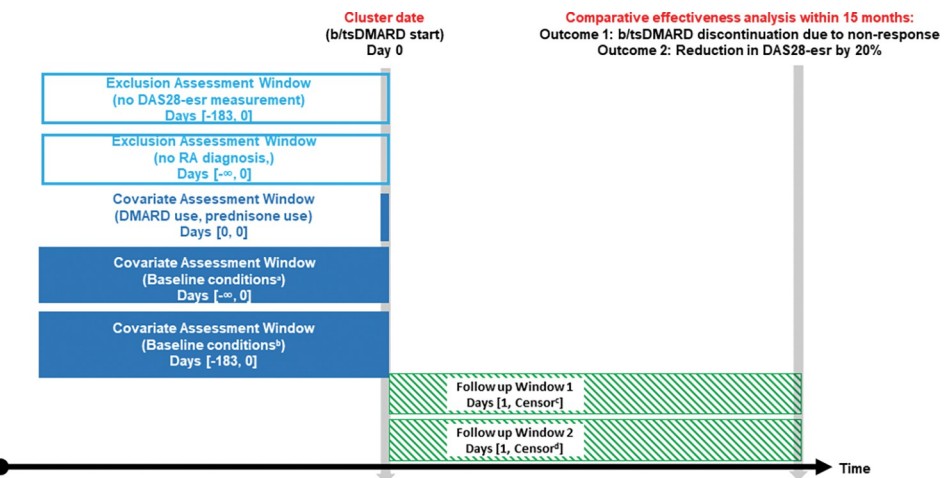

**Fig 1. Overview of the study composition.** RA: rheumatoid arthritis; DAS28-esr: rheumatoid arthritis disease activity score based on 28 joints and erythrocyte sedimentation rate; b/tsDMARD: biologic or targeted synthetic disease modifying antirheumatic drug.

## Outcome

We assessed 2 separate outcomes (i.e. non-response, treatment response) within a time window of 15 months. The outcome non-response was defined as discontinuation of the b/tsDMARD due to physician-recorded non-response (binary). The outcome treatment response was defined as a reduction in DAS28-esr by $\geq$20% (binary).

## Follow-up

When assessing non-response, we followed all patients from the cohort entry (i.e. b/tsDMARD start) until occurrence of treatment discontinuation due to non-response (i.e. the outcome), or censoring due to other treatment discontinuation, end of follow-up (15 months), or end of patient record, whichever happened first (Fig 1).

When assessing treatment response, we followed all patients from the cohort entry (i.e., b/tsDMARD start) until occurrence of 20% reduction in DAS28-esr (i.e. the outcome), or censoring due to any treatment discontinuation, end of follow-up (15 months), or end of patient record, whichever happened first (Fig 1).

## Clustering

We clustered patients at cohort entry (i.e., start of b/tsDMARD) using deep embedded clustering (DEC) [17] in combination with the AnyNets-Autoencoder, an adaptive deep adaptive neural network, which is especially designed to work with medical data with missing values [18]. The adaptive architecture outperformed classical feed-forward neural networks in combination with imputation methods (a naive approach to deal with missing values by replacing them with defaults such as e.g. zeros) [19]. DEC simultaneously learns feature representations and cluster assignments in a lower-dimensional space (latent space) using the adaptive deep autoencoder in order to reconstruct the input and iteratively optimizes a clustering objective [20]. Details about the algorithm and the architecture can be found in S2 Text and S1 Fig. The number of clusters is not decided by the algorithm and has to be given as hyperparameter. We evaluated the found clusters using the silhouette score, which measures the quality of the cluster assignment based on how similar data points are to their own cluster compared to other clusters. The score ranges from -1 to +1, where a high value denotes a well matched and separated clustering.

Since the aim of this study was to find patient groups that may respond differently to certain treatments, one clustering run yielding only a few clusters will not lead to robust results in comparative effectiveness analyses. Thus, taking into account the performance of clustering and sample size, we decided for 3, 4, and 5 clusters whose response to different treatments we were then able to compare. The clustering was implemented using Python in which we used the deep learning library Pytorch.

In post-hoc analyses to compare obtained clusters, k-means clustering without deep learning was performed. We did not obtain separated clusters (S2 to S7 Figs).

## Features

Features for patient clustering included patient demographics and life-style factors (e.g., physical activity, body mass index, smoking) that were considered time-invariant and thus captured using an ever before lookback window. Additional features such as other medication use (e.g., conventional synthetic [cs] DMARDs) were assessed at cohort entry. Laboratory markers (e.g., esr), other clinical feature (e.g., number of swollen/tender joints), and health measurement / disability scores (e.g., health assessment questionnaire) were assessed from a 6-month

lookback window and if not present considered as missing. Detailed information on used features and the level of missingness is available in S1 Table.

Additional clustering runs to further increase robustness of our results accounted for the recency of the last measurement of laboratory markers, disease activity, and health measurement / disability scores by adding the time since measurement (within 6 months) as an additional input feature. Thus, given the pre-defined number of 3, 4, and 5 clusters and having separate runs not accounting and accounting for the recency of the last clinical measurements, we will obtain a total of 24 clusters (i.e. (3+4+5)*2 = 24) which was a good number in terms of achieving robust results and manual cluster handling and result interpretation.

## Data analyses

**Descriptive analyses.** The workflow of performed steps following the clustering can be seen in Fig 2. We described the patient characteristics overall and visually inspected each cluster. Because findings from individual clusters may be chance findings and each cluster was similar in patient characteristics to at least one other cluster, we grouped clusters manually according to most distinct patient characteristics. The grouping was based upon visual comparison of patient characteristics. Most extreme characteristics (i.e. highest and lowest

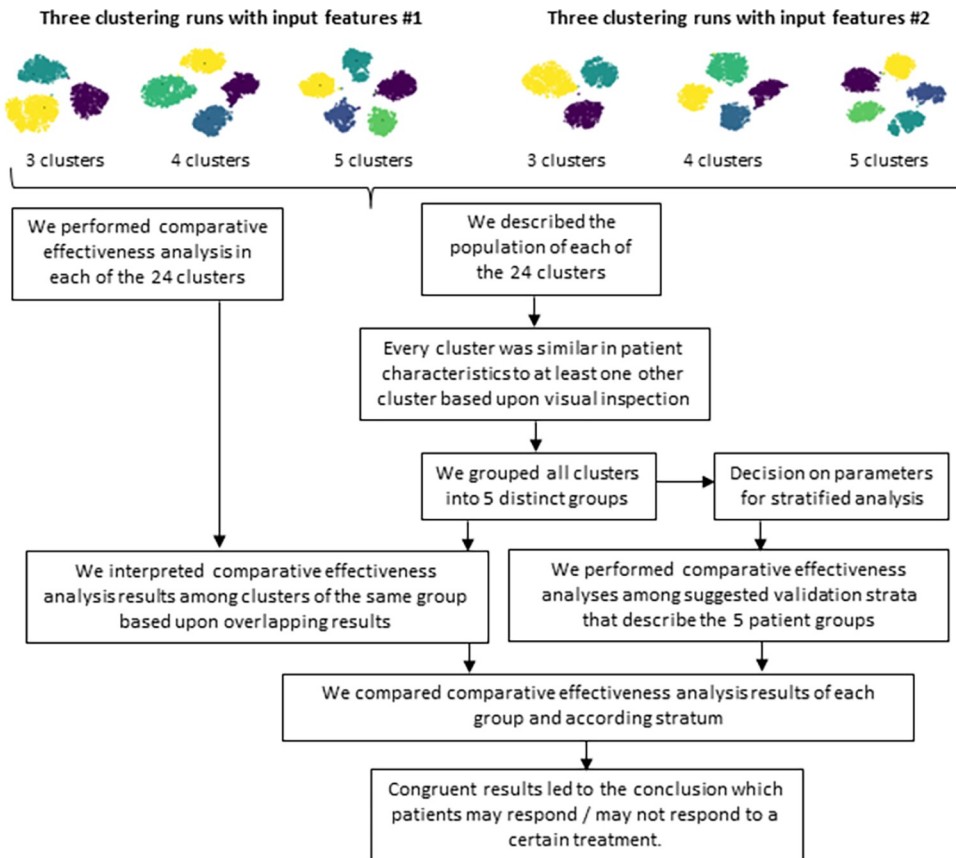

**Fig 2. Work flow of performed steps.** The top part depicts a 2D latent representation of all patients using t-distributed Stochastic Neighbor Embedding from deep embedded clustering with 3 clusters, 4 clusters, and 5 clusters in separate runs once accounting for the recency of the last clinical measurements (input features #2) and once not (input features #1). Different colors denote the different cluster assignments. Information on input features are available in S1 Table.

frequencies/values) were marked, and by "putting these together", groups were formed. An example with few clusters and patient characteristics is given in S3 Text. Significance testing of patient characteristics using ANOVA (Analysis of Variance) and subsequent Least Significance Difference tests was not feasible given the number of clusters and variables.

When clustering patients, we use the strength of machine learning techniques and include many features (i.e. patient characteristics). Using these results in clinical decision-making implies that all characteristics would need to be assessed at the moment of decision-making which is not feasible. Therefore, we further employ stratified analyses which make results applicable to clinical practice. Stratification means that patients are categorized according to few (most important) characteristics only and it is the preferred method to validate results obtained through clustering. In this study, we used the strength of clustering to set the basis to define the most distinct patient characteristics that set the different groups apart, which would not have been possible otherwise. Categories of these most distinct features were then derived as parameters for the stratified (validation) analyses (Fig 2).

**Comparative effectiveness analyses.**   In each of the 24 clusters, we performed comparative effectiveness analysis of b/tsDMARDs by using Cox proportional hazard regression analysis to estimate crude hazard ratios (HRs) with 95% confidence intervals (CIs) of treatment non-response, and treatment response separately. Because findings from individual clusters may be chance findings, we compared results among clusters belonging to the same group. Furthermore, to increase robustness of our reported findings, we only report significant results that were present in the maximum number of clusters of the same group (Fig 2).

**Validation.**   Finally, to validate obtained results, we repeated comparative effectiveness analyses in stratified analyses according to parameters derived in the descriptive analyses part. We estimated age- (and sex) adjusted HR with 95% CIs. We performed the description of patient clusters and comparative effectiveness analyses using SAS statistical software version 9.4 (NC, USA).

## Results

### Descriptive analyses

We identified 3516 patients with a first-time b/tsDMARD in SCQM between 1999 and 2018 (flow chart in S8 Fig). Selected patient characteristics of the overall population can be seen in Table 1. Patients had a mean age of 55.4 years and 76.2% of patients were women. The median RA duration of patients was 6.2 years, 69.3% were rheumatoid factor (RF) positive, the mean DAS28-esr was 4.3, and 66.6% of patients also used methotrexate at their first b/tsDMARD use.

Among 24 obtained clusters, the clusters differed in size and comprised a minimum of 362 and a maximum of 1481 patients. Most distinct patient characteristics between clusters were sex, seropositivity, csDMARD use, prednisone use, measures of disease activity and pain, and disease duration. Visual inspection of patient characteristics of all 24 clusters led to their grouping into 5 distinct groups. Patient characteristics of each cluster (already grouped into 5 groups) can be seen in S2 to S6 Tables. The first group comprised clusters with highest frequencies of individual csDMARD and prednisone use ($\geq$75%, $\geq$31%, $\geq$19%, and $\geq$47% of patients used methotrexate, leflunomide, and sulfasalazine, and prednisone, respectively, at b/tsDMARD initiation), and with a tendency towards seropositivity and no family history of rheumatic diseases. Parameters derived for the validation analysis of this group were the use of $\geq$2 csDMARDs and prednisone. The second group comprised clusters of a majority of men (>99%) which further displayed the highest frequency of smoking and mean body mass index (both assessed from an ever before lookback window). The parameter derived for the

**Table 1. Selected patient characteristics at first-time b/tsDMARD.**

| Patient characteristic at cohort entry | Study population n = 3516 |
|---|---|
| **Mean Age [years] (SD)** | 55.4 (13.6) |
| **Women** | 2679 (76.2%) |
| **Men** | 837 (23.8%) |
| **Mean BMI (SD)** | 25.5 (5.0) |
| **Median RA duration[a](IQR)** | 6.2 (2.5–13.5) |
| **No family history of rheumatic diseases[b]** | 1589 (45.2%) |
| **Family history of rheumatic diseases[b]** | 788 (22.4%) |
| **Missing information on family history** | 1139 (32.4%) |
| **Rheumatoid factor negative** | 933 (26.5%) |
| **Rheumatoid factor positive** | 2435 (69.3%) |
| **Missing rheumatoid factor information** | 148 (4.2%) |
| **ACPA negative** | 897 (25.5%) |
| **ACPA positive** | 1656 (47.1%) |
| **Missing ACPA information** | 963 (27.4%) |
| **Mean DAS28-esr score (SD)** | 4.3 (1.2) |
| **Mean pain level [VAS] (SD)** | 4.8 (2.8) |
| **Missing pain level information** | 399 (11.3%) |
| **Mean HAQ score (SD)** | 1.0 (0.7) |
| **Missing HAQ score information** | 430 (12.2%) |
| **Mean SF 12 physical component score (SD)** | 34.9 (10.0) |
| **Mean SF 12 mental component score (SD)** | 45.6 (11.9) |
| **Missing SF 12 information** | 726 (20.6%) |
| **Methotrexate use** | 2342 (66.6%) |
| **Mean duration of methotrexate use (SD)** | 3.4 (1.8) |
| **Leflunomide use** | 925 (26.3%) |
| **Mean duration of leflunomide use (SD)** | 2.3 (1.2) |
| **Sulfasalazin use** | 652 (18.5%) |
| **Mean duration of sulfasalazin use (SD)** | 3.6 (1.9) |
| **Prednisone use** | 1537 (43.7%) |
| **Mean duration of prednisone use (SD)** | 2.6 (1.1) |

ACPA: Anti-citrullinated protein antibodies; DAS: disease activity score; esr: erythrocyte sedimentation rate; HAQ: health assessment questionnaire; IQR: interquartile range; RA: rheumatoid arthritis; SD: Standard deviation; SF: Short form (health survey)

[a] RA duration assessed from diagnosis until cohort entry, if diagnosis date not available assessed RA duration from first symptoms minus 1 year

[b] family anamnesis includes rheumatoid arthritis, ankylosing spondylitis, psoriasis, psoriatic arthritis, chronic inflammatory bowel disease, and other spondyloarthropathies (e.g. reactive arthritis)

validation analysis of this group was being male. The third group comprised clusters of mainly seronegative patients (>99% RF negative) with lowest frequency of prednisone use (<40%) and a tendency towards a higher proportion of women (>76%). Parameters derived for the validation analysis of this group were being seronegative and without prednisone use, and in a separate analysis additionally being female. The fourth group comprised clusters of mainly seropositive patients (>80% RF positive), with a rather high disease burden (mean DAS28-esr >4.4, mean HAQ score >1.2, mean VAS >4.7 [out of 10], and activity of disease >4.9 [out of 10]), long disease duration (median of >8 years), and a tendency towards being female

(≥77%). Parameters derived for the validation analysis of this group were being seropositive, with a DAS28-esr score >5.1, a HAQ score >1.5, pain levels >6, or activity of disease levels >6, and disease duration >8 years, and in a separate analysis additionally being female. The fifth group comprised clusters with a rather low disease burden (mean DAS28-esr ≤4.2, mean HAQ score ≤0.8, mean VAS ≤3.9 [out of 10], and activity of disease ≤4.2 [out of 10]) and with a tendency towards seropositivity and a higher proportion of women. Parameters derived for the validation analysis of this group were having a disease activity of a DAS28-esr score ≤3.2, a HAQ score <0.7, pain levels of <4 (of maximum 10), or activity of disease levels of <4 (of maximum 10), and in a separate analysis additionally being female and seropositive.

## Comparative effectiveness analyses

S4 Text depicts comparative effectiveness results obtained per cluster (grouped into the 5 distinct groups for ease of comparison). To increase robustness of our results, we only reported significant results that were observed in the majority of clusters per group. Clusters in the first group (i.e., high frequency of use of csDMARDs and prednisone) responded well to tocilizumab (significant HRs of 2.55–2.67, count of outcomes: 6–11, depending on the cluster). Clusters in the second group (i.e., men) also responded well to tocilizumab (significant HRs of 6.78–8.54, count of outcomes: 5–7, depending on the cluster). Clusters in the third group (i.e., seronegative patients with a tendency towards low use of prednisone and a higher proportion of women) as well as clusters in the fourth group (i.e., mainly seropositive women with high disease burden and long duration) had a high risk of non-response with golimumab (significant HRs of 2.10–2.53 and of 2.45–4.54, count of outcomes: 19–24 and 8–12, respectively, depending on the clusters). Clusters in the fifth group (i.e., patients with low disease burden and with a tendency towards seropositivity and a higher proportion of women) responded well to golimumab (significant HRs of 2.56–3.15, count of outcomes: 5–9, depending on the cluster) and tocilizumab (significant HRs of 3.39–6.43, depending on the cluster).

## Validation

Comparative effectiveness results in each group were validated by stratified analyses according to most distinct patient characteristics derived from each group to make results applicable to clinical practice. Tables 2–6 show comparative effectiveness results in each derived patient strata. First, among 728 patients with ≥2 csDMARDs and use of prednisone at index date, we confirmed the good response to tocilizumab with an age and sex adjusted HR of 5.46 (95% CI 1.76–16.94) when compared to adalimumab (Table 2). Furthermore, we observed a significant good response to golimumab and tofacitinib, and an increased risk of non-response to golimumab. Second, among 837 men, the good response to tocilizumab was confirmed with an age adjusted HR of 8.44 (95% CI 3.43–20.74) [Table 3]. Moreover, we observed a significant good response to golimumab and tofacitinib, but also a significant non-response to golimumab. Third, among 590 seronegative patients without use of prednisone, an increased risk of non-response with golimumab was not confirmed (age and sex adjusted HR of 1.98, 95% CI 0.94–4.19) [Table 4]. However, in a more specific population containing only women (n = 459), we observed a significant result of non-response with golimumab (age adjusted HR of 2.36, 95% CI 1.03–5.40). Fourth, among both 717 patients and 587 women with seropositivity, high disease burden and disease duration, we confirmed a high risk of non-response with golimumab with adjusted HR of 3.75 (95% CI 1.54–9.12) and of 5.27 (95% CI 2.10–13.21), respectively [Table 5]. Fifth, among both 2466 patients with low disease burden and 1313 seropositive women with low disease burden, we confirmed a good response to golimumab with adjusted HRs of 2.72 (95% CI 1.63–4.57) and of 2.39 (95% CI 1.06–5.41), respectively (Table 6). In these

**Table 2. Comparative effectiveness analyses in strata of patients with at least 2 csDMARDs and use of prednisone.**

| | Treatment disc. due to non-response | Crude HR (95% CI) | Age and sex adjusted HR (95% CI) | 20% DAS28-esr reduction | Crude HR (95% CI) | Age and sex adjusted HR (95% CI) |
|---|---|---|---|---|---|---|
| TNF-inhibitor | 83 | Ref 1.00[a] | Ref 1.00[a] | 37 | Ref 1.00[a] | Ref 1.00[a] |
| Adalimumab | 33 | Ref 1.00[b] | Ref 1.00[b] | 8 | Ref 1.00[b] | Ref 1.00[b] |
| Certolizumab | 3 | NA | NA | 2 | NA | NA |
| Etanercept | 23 | 0.81 (0.48–1.39) | 0.84 (0.49–1.43) | 13 | 1.97 (0.82–4.76) | 2.01 (0.83–4.86) |
| Golimumab | 13 | 3.13 (1.65–5.96) | 3.47 (1.81–6.68) | 7 | 7.75 (2.8–21.46) | 8.27 (2.96–23.11) |
| Infliximab | 11 | 0.60 (0.31–1.20) | 0.61 (0.31–1.20) | 7 | 1.83 (0.66–5.05) | 1.83 (0.66–5.04) |
| Non-TNF-inh. | 12 | 0.81 (0.44–1.49) | 0.79 (0.43–1.47) | 10 | 1.62 (0.80–3.26) | 1.60 (0.78–3.26) |
| Abatacept | 5 | NA | 0.75 (0.29–1.95) | 3 | NA | NA |
| Rituximab | 2 | NA | NA | 2 | NA | NA |
| Tocilizumab | 5 | 1.12 (0.44–2.87) | 1.17 (0.45–3.01) | 5 | 5.36 (1.75–16.44) | 5.46 (1.76–16.94) |
| JAK-inhibitor | 4 | NA | NA | 6 | 6.33 (2.67–15.05) | 6.09 (2.51–14.76) |
| Baricitinib | 0 | NA | NA | 1 | NA | NA |
| Tofacitinib | 4 | NA | NA | 5 | 10.22 (3.33–31.30) | 9.37 (3.02–29.11) |

CI: confidence interval; Disc.: discontinuation; esr: erythrocyte sedimentation rate; HR: hazard radio; inh.: inhibitor; JAK: janus kinase; TNF: tumor necrosis factor alpha

[a] The reference group in class effect analyses was TNF inhibitors

[b] The reference group in individual b/tsDMARD analyses was adalimumab.

strata, we further confirmed a good response to tocilizumab with adjusted HRs of 3.64 (95% CI 2.04–6.49) and of 2.98 (95% CI 1.31–6.74), respectively (Table 6). Furthermore, we observed significant good responses to tofacitinib, and a significant non-response to golimumab.

## Discussion

Through deep embedded clustering, we obtained 5 distinct RA patient groups which turned out to respond differently to certain b/tsDMARDs. Results were validated using stratified analyses with fewer, most important features to make results applicable for clinical practice.

A strength of this study is that we used deep learning-based clustering that was shown to perform better than regular clustering [21]. We included temporal trends of characteristics by adding the time since the last measurement as a feature into the second clustering run. Yet, incorporating so called attentions into our clustering algorithm leading towards a transformer architecture may have been even more powerful [22]. While the use of clustering in this study also implied an exploratory approach (since we did not know whether obtained groups may prove useful), manyfold exploratory comparative effectiveness analyses among strata of various combinations of patient characteristics will be impracticable. A further strength includes the use of several clustering runs that obtained 2 dozen different clusters which we grouped according to similar patient characteristics. We admit that this process made the study more complex and potentially less reproducible because the visual inspection of clusters to form distinct groups may lead to different grouping depending on who is inspecting the clusters. A continuation of this work could include the automation of grouping of clusters through learning of a meta-classifier for example. However, such a program does not exist yet. While

**Table 3. Comparative effectiveness analyses in men stratum.**

| | Treatment disc. due to non-response | Crude HR (95% CI) | Age and sex adjusted HR (95% CI) | 20% DAS28-esr reduction | Crude HR (95% CI) | Age and sex adjusted HR (95% CI) |
|---|---|---|---|---|---|---|
| TNF-inhibitor | 87 | Ref 1.00[a] | Ref 1.00[a] | 53 | Ref 1.00[a] | Ref 1.00[a] |
| Adalimumab | 37 | Ref 1.00[b] | Ref 1.00[b] | 16 | Ref 1.00[b] | Ref 1.00[b] |
| Certolizumab | 1 | NA | NA | 1 | NA | NA |
| Etanercept | 25 | 0.81 (0.49–1.35) | 0.81 (0.49–1.35) | 18 | 1.45 (0.74–2.84) | 1.44 (0.74–2.83) |
| Golimumab | 13 | 2.24 (1.19–4.21) | 2.22 (1.18–4.18) | 8 | 2.96 (1.27–6.92) | 2.93 (1.25–6.88) |
| Infliximab | 11 | 0.54 (0.28–1.06) | 0.54 (0.27–1.05) | 10 | 1.23 (0.56–2.71) | 1.22 (0.55–2.70) |
| Non-TNF-inh. | 13 | 1.32 (0.74–2.37) | 1.35 (0.75–2.43) | 11 | 2.04 (1.06–3.90) | 2.12 (1.10–4.08) |
| Abatacept | 7 | 1.26 (0.56–2.82) | 1.27 (0.57–2.87) | 0 | NA | NA |
| Rituximab | 2 | NA | NA | 4 | NA | NA |
| Tocilizumab | 4 | NA | NA | 7 | 8.59 (3.52–20.94) | 8.44 (3.43–20.74) |
| JAK-inhibitor | 3 | NA | NA | 6 | 3.77 (1.62–8.78) | 3.94 (1.68–9.22) |
| Baricitinib | 0 | NA | NA | 0 | NA | NA |
| Tofacitinib | 3 | NA | NA | 6 | 5.03 (1.97–12.86) | 5.09 (1.98–13.06) |

CI: confidence interval; Disc.: discontinuation; esr: erythrocyte sedimentation rate; HR: hazard radio; inh.: inhibitor; JAK: janus kinase; TNF: tumor necrosis factor alpha

[a] The reference group in class effect analyses was TNF inhibitors

[b] The reference group in individual b/tsDMARD analyses was adalimumab.

grouping of clusters may take away some information on clusters that machine learning techniques provided, results also need to be robust. The comparison of comparative effectiveness results among groups of similar clusters made our results less prone to chance findings. Having assessed similar but not identical clusters can be considered like a type of re-sampling. The overlap between clusters was indeed desired and needed to obtain robust results insensitive to minor variation. This was important because sample size of individual clusters was small. Finally, since the particular b/tsDMARD agent was not a feature in the clustering and a study in SCQM suggested no changes in patient characteristics at bDMARD initiation over time, [23] our cluster results have not been biased from time trends. Furthermore, we can also likely rule out bias from patient and physician expectations because there is no reason for a differentially distributed expectation between individual b/tsDMARD treatments since EULAR RA treatment guidance does not commend one over another [2].

While machine learning is a powerful tool to detect patterns not visible to the human eye, the use of obtained clusters in clinical practice is not feasible because clusters display an abundance of features. In clinical practice, we need simple criteria–if possible–by which to categorize patients in whom treatment response may differ. Since stratification is the preferred method to validate results obtained in patient clusters, we attempted this in this study. However, also stratification was a manual process subject to subjectivity. And since sample size was small, we were limited in the number of strata. Thus, maybe we simplified the strata too much. However, more strata mean more multiple testing. Thus, it was a trade-off to have meaningful strata according to most distinct patient characteristics while keeping the number of strata to a minimum. Moreover, despite small sample size, we observed congruent findings between comparative effectiveness analyses among clusters and validation strata. Yet, also the process to derive most important characteristics may be improved and automatized in future studies

**Table 4. Comparative effectiveness analyses in strata of seronegative patients without use of prednisone, and more specifically, among women only.**

| | Treatment disc. due to non-response | Crude HR (95% CI) | Age (and sex) adjusted HR (95% CI) | 20% DAS28-esr reduction | Crude HR (95% CI) | Age (and sex) adjusted HR (95% CI) |
|---|---|---|---|---|---|---|
| **Seronegative patients without use of prednisone** | | | | | | |
| **TNF-inhibitor** | 70 | Ref 1.00[a] | Ref 1.00[a] | 45 | Ref 1.00[a] | Ref 1.00[a] |
| **Adalimumab** | 23 | Ref 1.00[b] | Ref 1.00[b] | 14 | Ref 1.00[b] | Ref 1.00[b] |
| **Certolizumab** | 3 | NA | NA | 1 | NA | NA |
| **Etanercept** | 25 | 0.91 (0.52–1.61) | 1.00 (0.57–1.78) | 20 | 1.21 (0.61–2.40) | 1.23 (0.62–2.46) |
| **Golimumab** | 10 | 1.90 (0.90–3.99) | 1.98 (0.94–4.19) | 3 | NA | NA |
| **Infliximab** | 9 | 0.82 (0.38–1.77) | 0.82 (0.38–1.78) | 7 | 1.02 (0.41–2.52) | 1.03 (0.41–2.54) |
| **Non-TNF-inh.** | 8 | 1.10 (0.53–2.29) | 0.96 (0.46–2.01) | 6 | 1.33 (0.57–3.11) | 1.37 (0.58–3.23) |
| **Abatacept** | 6 | 1.86 (0.76–4.56) | 1.69 (0.69–4.17) | 2 | NA | NA |
| **Rituximab** | 1 | NA | NA | 1 | NA | NA |
| **Tocilizumab** | 1 | NA | NA | 3 | NA | NA |
| **JAK-inhibitor** | 1 | NA | NA | 3 | NA | NA |
| **Baricitinib** | 0 | NA | NA | 0 | NA | NA |
| **Tofacitinib** | 1 | NA | NA | 3 | NA | NA |
| **Seronegative women without use of prednisone** | | | | | | |
| **TNF-inhibitor** | 56 | Ref 1.00[a] | Ref 1.00[a] | 34 | Ref 1.00[a] | Ref 1.00[a] |
| **Adalimumab** | 15 | Ref 1.00[b] | Ref 1.00[b] | 10 | Ref 1.00[b] | Ref 1.00[b] |
| **Certolizumab** | 3 | NA | NA | 1 | NA | NA |
| **Etanercept** | 20 | 1.02 (0.52–2.00) | 1.11 (0.57–2.18) | 14 | 1.07 (0.48–2.41) | 1.08 (0.48–2.43) |
| **Golimumab** | 9 | 2.25 (0.99–5.14) | 2.36 (1.03–5.40) | 2 | NA | NA |
| **Infliximab** | 9 | 1.20 (0.52–2.73) | 1.17 (0.51–2.68) | 7 | 1.34 (0.51–3.53) | 1.34 (0.51–3.53) |
| **Non-TNF-inh.** | 7 | 1.02 (0.46–2.24) | 0.93 (0.42–2.04) | 5 | 1.21 (0.47–3.10) | 1.21 (0.47–3.10) |
| **Abatacept** | 6 | 2.29 (0.89–5.90) | 2.23 (0.87–5.76) | 2 | NA | NA |
| **Rituximab** | 0 | NA | NA | 1 | NA | NA |
| **Tocilizumab** | 1 | NA | NA | 2 | NA | NA |
| **JAK-inhibitor** | 1 | NA | NA | 1 | NA | NA |
| **Baricitinib** | 0 | NA | NA | 0 | NA | NA |
| **Tofacitinib** | 1 | NA | NA | 1 | NA | NA |

CI: confidence interval; Disc.: discontinuation; esr: erythrocyte sedimentation rate; HR: hazard radio; inh.: inhibitor; JAK: janus kinase; TNF: tumor necrosis factor alpha

[a] The reference group in class effect analyses was TNF inhibitors

[b] The reference group in individual b/tsDMARD analyses was adalimumab.

to ensure reproducibility. A further limitation of our study is that we did not have an external cohort at hand to further validate our results externally. Thus, caution is warranted when using our findings in clinical practice. Furthermore, to assess a change in DAS is not a stringent outcome and can only be considered an approximation of treatment response. However, it allowed us to increase sample size without which an investigation in this direction would not have been possible. Finally, we were not able to adjust our analyses for confounding variables like obesity because of the small sample size. However, crude and age/sex adjusted HR were identical for most of our results. Thus, we do not expect an influence on our results from other measured or non-measured confounding. While time-varying confounder adjustment may be

**Table 5. Comparative effectiveness analyses in the stratum of patients with seropositivity, with high disease activity and disease duration, and additionally among women only.**

| | Treatment disc. due to non-response | Crude HR (95% CI) | Age (and sex) adjusted HR (95% CI) | 20% DAS28-esr reduction | Crude HR (95% CI) | Age (and sex) adjusted HR (95% CI) |
|---|---|---|---|---|---|---|
| **Seropositive patients with rather high disease activity and duration** | | | | | | |
| TNF-inhibitor | 62 | Ref 1.00[a] | Ref 1.00[a] | 44 | Ref 1.00[a] | Ref 1.00[a] |
| **Adalimumab** | 26 | Ref 1.00[b] | Ref 1.00[b] | 18 | Ref 1.00[b] | Ref 1.00[b] |
| **Certolizumab** | 0 | NA | NA | 0 | NA | NA |
| **Etanercept** | 18 | 0.59 (0.32–1.08) | 0.59 (0.32–1.08) | 16 | 0.86 (0.44–1.68) | 0.83 (0.42–1.63) |
| **Golimumab** | 6 | 3.74 (1.54–9.08) | 3.75 (1.54–9.12) | 2 | NA | NA |
| **Infliximab** | 12 | 0.68 (0.34–1.35) | 0.68 (0.34–1.35) | 8 | 0.85 (0.37–1.94) | 0.83 (0.36–1.90) |
| Non-TNF-inh. | 6 | 0.87 (0.38–2.00) | 0.87 (0.37–2.01) | 5 | 1.13 (0.45–2.84) | 1.10 (0.44–2.78) |
| **Abatacept** | 3 | NA | NA | 1 | NA | NA |
| **Rituximab** | 1 | NA | NA | 2 | NA | NA |
| **Tocilizumab** | 2 | NA | NA | 2 | NA | NA |
| **JAK-inhibitor** | 0 | NA | NA | 4 | NA | NA |
| **Baricitinib** | 0 | NA | NA | 3 | NA | NA |
| **Tofacitinib** | 0 | NA | NA | 1 | NA | NA |
| **Seropositive women with rather high disease activity and duration** | | | | | | |
| TNF-inhibitor | 49 | Ref 1.000[a] | Ref 1.000[a] | 36 | Ref 1.000[a] | Ref 1.000[a] |
| **Adalimumab** | 19 | Ref 1.00[b] | Ref 1.00[b] | 15 | Ref 1.00[b] | Ref 1.00[b] |
| **Certolizumab** | 0 | NA | NA | 0 | NA | NA |
| **Etanercept** | 15 | 0.66 (0.34–1.30) | 0.65 (0.33–1.29) | 13 | 0.79 (0.37–1.65) | 0.75 (0.35–1.58) |
| **Golimumab** | 6 | 5.24 (2.09–13.14) | 5.27 (2.10–13.21) | 1 | NA | NA |
| **Infliximab** | 9 | 0.64 (0.29–1.41) | 0.64 (0.29–1.40) | 7 | 0.80 (0.33–1.97) | 0.76 (0.31–1.88) |
| Non-TNF-inh. | 4 | NA | NA | 2 | NA | NA |
| **Abatacept** | 2 | NA | NA | 1 | NA | NA |
| **Rituximab** | 1 | NA | NA | 1 | NA | NA |
| **Tocilizumab** | 1 | NA | NA | 0 | NA | NA |
| **JAK-inhibitor** | 0 | NA | NA | 3 | NA | NA |
| **Baricitinib** | 0 | NA | NA | 3 | NA | NA |
| **Tofacitinib** | 0 | NA | NA | 0 | NA | NA |

CI: confidence interval; Disc.: discontinuation; esr: erythrocyte sedimentation rate; HR: hazard radio; inh.: inhibitor; JAK: janus kinase; TNF: tumor necrosis factor alpha

[a] The reference group in class effect analyses was TNF inhibitors

[b] The reference group in individual b/tsDMARD analyses was adalimumab.

desirable in comparative effectiveness analyses in rheumatologic conditions, markers of pain, disability, and disease activity often lie on the causal pathway between exposure and outcome and should therefore not be included in the model.

Obtained clusters in three recent studies using k-means clustering to identify RA phenotypes were mainly driven by disease activity and comorbidities, sex and comorbidities, and disease activity and health care costs, respectively [11–13]. In our study, we did not assess comorbidities or health care costs, but we can confirm the importance of disease activity and sex, and we additionally identified seropositivity, disease duration, and other RA medication

**Table 6. Comparative effectiveness analyses in the stratum of patients with low disease activity, and additionally among women with seropositivity only.**

| | Treatment disc. due to non-response | Crude HR (95% CI) | Age and sex adjusted HR (95% CI) | 20% DAS28-esr reduction | Crude HR (95% CI) | Age and sex adjusted HR (95% CI) |
|---|---|---|---|---|---|---|
| **Patients with low disease activity** | | | | | | |
| **TNF-inhibitor** | 220 | Ref 1.00[a] | Ref 1.00[a] | 140 | Ref 1.00[a] | Ref 1.00[a] |
| **Adalimumab** | 81 | Ref 1.00[b] | Ref 1.00[b] | 42 | Ref 1.00[b] | Ref 1.00[b] |
| **Certolizumab** | 7 | 1.41 (0.65–3.05) | 1.41 (0.65–3.05) | 6 | 2.36 (1.00–5.56) | 2.31 (0.98–5.43) |
| **Etanercept** | 59 | 0.71 (0.51–0.99) | 0.71 (0.51–0.99) | 51 | 1.21 (0.81–1.83) | 1.23 (0.82–1.86) |
| **Golimumab** | 42 | 2.51 (1.73–3.64) | 2.51 (1.73–3.65) | 22 | 2.72 (1.62–4.56) | 2.72 (1.63–4.57) |
| **Infliximab** | 31 | 0.69 (0.46–1.05) | 0.69 (0.46–1.05) | 19 | 0.92 (0.53–1.57) | 0.91 (0.53–1.57) |
| **Non-TNF-inh.** | 37 | 1.17 (0.83–1.66) | 1.18 (0.83–1.67) | 29 | 1.58 (1.06–2.36) | 1.63 (1.09–2.45) |
| **Abatacept** | 21 | 1.40 (0.86–2.26) | 1.39 (0.86–2.26) | 8 | 1.11 (0.52–2.37) | 1.13 (0.53–2.42) |
| **Rituximab** | 3 | NA | NA | 5 | 1.50 (0.60–3.80) | 1.57 (0.62–3.98) |
| **Tocilizumab** | 13 | 1.16 (0.65–2.09) | 1.16 (0.65–2.09) | 16 | 3.49 (1.96–6.21) | 3.64 (2.04–6.49) |
| **JAK-inhibitor** | 6 | 0.80 (0.36–1.81) | 0.80 (0.36–1.81) | 13 | 2.97 (1.68–5.25) | 3.00 (1.69–5.31) |
| **Baricitinib** | 0 | NA | NA | 2 | NA | NA |
| **Tofacitinib** | 6 | 0.80 (0.35–1.84) | 0.80 (0.35–1.84) | 11 | 3.23 (1.67–6.28) | 3.22 (1.65–6.27) |
| **Seropositive women with low disease activity** | | | | | | |
| **TNF-inhibitor** | 102 | Ref 1.00[a] | Ref 1.00[a] | 69 | Ref 1.00[a] | Ref 1.00[a] |
| **Adalimumab** | 37 | Ref 1.00[b] | Ref 1.00[b] | 21 | Ref 1.00[b] | Ref 1.00[b] |
| **Certolizumab** | 4 | NA | NA | 4 | NA | NA |
| **Etanercept** | 27 | 0.66 (0.40–1.09) | 0.67 (0.41–1.10) | 24 | 1.05 (0.59–1.89) | 1.06 (0.59–1.90) |
| **Golimumab** | 14 | 2.07 (1.12–3.82) | 2.08 (1.12–3.85) | 8 | 2.38 (1.05–5.38) | 2.39 (1.06–5.41) |
| **Infliximab** | 20 | 0.88 (0.51–1.51) | 0.88 (0.51–1.51) | 12 | 1.06 (0.52–2.16) | 1.07 (0.53–2.18) |
| **Non-TNF-inh.** | 20 | 1.29 (0.80–2.08) | 1.33 (0.82–2.15) | 13 | 1.35 (0.75–2.45) | 1.40 (0.77–2.55) |
| **Abatacept** | 9 | 1.24 (0.60–2.58) | 1.28 (0.61–2.67) | 5 | 1.36 (0.51–3.62) | 1.41 (0.53–3.78) |
| **Rituximab** | 2 | NA | NA | 0 | NA | NA |
| **Tocilizumab** | 9 | 1.54 (0.74–3.19) | 1.56 (0.75–3.24) | 8 | 2.93 (1.30–6.63) | 2.98 (1.31–6.74) |
| **JAK-inhibitor** | 3 | NA | NA | 4 | NA | NA |
| **Baricitinib** | 0 | NA | NA | 2 | NA | NA |
| **Tofacitinib** | 3 | NA | NA | 2 | NA | NA |

CI: confidence interval; Disc.: discontinuation; esr: erythrocyte sedimentation rate; HR: hazard radio; inh.: inhibitor; JAK: janus kinase; TNF: tumor necrosis factor alpha

[a] The reference group in class effect analyses was TNF inhibitors

[b] The reference group in individual b/tsDMARD analyses was adalimumab.

use at b/tsDMARD start as important variables for differentiating between clusters. It seems that the RA population in the cited studies were rather diverse and not linked to RA treatment, whereas we clustered our patients at b/tsDMARD start. However, despite using different clustering techniques and patient populations, the overlap between results is promising. Our post-hoc approach using k-means clustering did not yield separate clusters which suggests that deep learned clusters may be superior to k-means clusters when working with missing data which is usually the case in health care data.

Epidemiologic studies to date have identified predictors of treatment response (or non-response) to certain b/tsDMARDs [24–28]. We compared our results with those because there

are no studies that identified response to b/tsDMARDs among RA phenotypes that consist of more than one trait. Observed non-response to golimumab among seronegative women without use of prednisone as well as seropositive patients with higher disease burden and duration may suggest to avoid this treatment in these patients. However, no other significant findings were identified in these strata to help with treatment choice. Thus, adalimumab, the reference treatment may be a suggestion; especially since a study which assessed patient characteristics associated with treatment response to adalimumab suggested also that concomitant csDMARD use as well as a high disease burden was predictive of treatment response [24].

A study assessing drug survival of golimumab within 2 years was not able to identify predictors of response to golimumab [25]. In our study, 3 strata (patients with ≥2 csDMARDs and prednisone use at b/tsDMARD initiation, male patients, as well as patients with a lower disease burden) yielded a good response to golimumab. Yet, it is likely that response was not sufficient to persist on golimumab therapy given the equally increased risk of treatment discontinuation with golimumab. This may have been due to our outcome proxy for good response, which was defined as a ≥20% reduction in DAS28-esr which only requires small improvement but yielded a sufficient number of outcomes to perform analyses.

The same 3 strata (patients with ≥2 csDMARDs and prednisone use at b/tsDMARD initiation, male patients, as well as patients with a lower disease burden) further yielded a good response to tofacitinib and tocilizumab. Our results are not consistent with another study suggesting that a higher disease burden is a predictor of treatment response to tocilizumab, however the outcome definition was different and the response was assessed after only 6 months [26]. Nonetheless, attention should be paid to tofacitinib which was recently shown to have an early effect on pain reduction [29,30] which may work most efficiently in these 3 strata (i.e. patients with ≥2 csDMARDs and prednisone use at b/tsDMARD initiation, male patients, as well as patients with a lower disease burden).

This is the first study to combine machine learning with comparative effectiveness analyses. Thereby, we hope to inspire other investigators to repeat our approach or to think of other good ways how to personalize treatments for RA patients by using all methods available (i.e., bridging machine learning with conventional statistical methods).

## Conclusion

This study used deep learning to suggest RA patient groups which responded differently to certain b/tsDMARDs. Results were validated through stratified analyses according to few most distinct patient characteristics for application in clinical practice. Our results suggests optimal first-line b/tsDMARD use in certain patient groups which is a step forward towards personalizing treatment in RA patients. However, further research in other cohorts is needed to verify our results.

## Supporting information

**S1 Text. More information on Swiss Clinical Quality Management in Rheumatic Diseases (SCQM) registry.**
(DOC)

**S2 Text. Clustering—Deep Neural Network Architecture and Training.**
(DOC)

**S3 Text. Example of approach how cluster groups were formed.**
(DOC)

**S4 Text. Comparative effectiveness analyses in each of the 24 clusters.**
(DOC)

**S1 Fig. Scheme of the adaptive deep neural network architecture *AnyNet-Autoencoder* with deep embedded clustering.**
(TIF)

**S2 Fig. Results of post hoc K-means clustering using specification #1 requesting 3 clusters.**
(TIF)

**S3 Fig. Results of post hoc K-means clustering using specification #1 requesting 4 clusters.**
(TIF)

**S4 Fig. Results of post hoc K-means clustering using specification #1 requesting 5 clusters.**
(TIF)

**S5 Fig. Results of post hoc K-means clustering using specification #2 requesting 3 clusters.**
(TIF)

**S6 Fig. Results of post hoc K-means clustering using specification #2 requesting 4 clusters.**
(TIF)

**S7 Fig. Results of post hoc K-means clustering using specification #2 requesting 5 clusters.**
(TIF)

**S8 Fig. Flowchart of the study population.**
(TIF)

**S1 Table. Features used for clustering at cohort entry including variable type, missingness, lookback window, and clustering run.**
(DOC)

**S2 Table. Clusters with high use of conventional synthetic DMARDs and prednisone.**
(DOC)

**S3 Table. Clusters of men.**
(DOC)

**S4 Table. Clusters of seronegative patients with a tendency towards low use of prednisone and a higher proportion of women.**
(DOC)

**S5 Table. Clusters of mainly seropositive patients with high RA disease burden and long RA disease duration, and with a tendency towards a higher proportion of women.**
(DOC)

**S6 Table. Clusters of patients with a rather low RA disease burden and with a tendency towards seropositivity and a higher proportion of women.**
(DOC)

## Acknowledgments

We thank all patients and rheumatologists contributing to the SCQM registry, as well as the entire SCQM staff. A list of rheumatology offices and hospitals which contribute to the SCQM registry and a list of supporters of SCQM can be found at http://www.scqm.ch.The professorship of Andrea M Burden is partially supported by PharmaSuisse and the ETH Foundation.

## Author Contributions

**Conceptualization:** Thomas Hügle, Theresa Burkard.

**Data curation:** Andrea M. Burden, Thomas Hügle.

**Formal analysis:** Maria Kalweit, Theresa Burkard.

**Investigation:** Maria Kalweit, Andrea M. Burden, Thomas Hügle, Theresa Burkard.

**Methodology:** Andrea M. Burden, Joschka Boedecker, Thomas Hügle, Theresa Burkard.

**Supervision:** Joschka Boedecker, Theresa Burkard.

**Validation:** Theresa Burkard.

**Visualization:** Theresa Burkard.

**Writing – original draft:** Maria Kalweit, Thomas Hügle, Theresa Burkard.

**Writing – review & editing:** Maria Kalweit, Andrea M. Burden, Joschka Boedecker, Thomas Hügle, Theresa Burkard.

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
