## [Decision Letter · Decision Letter 0]

21 Mar 2022

Dear Prof. Dr. Burden,

Thank you very much for submitting your manuscript "Patient Groups in Rheumatoid Arthritis Identified by Deep Learning Respond Differently to Biologic or Targeted Synthetic DMARDs" for consideration at PLOS Computational Biology.

As with all papers reviewed by the journal, your manuscript was reviewed by members of the editorial board and by several independent reviewers. In light of the reviews (below this email), we would like to invite the resubmission of a significantly-revised version that takes into account the reviewers' comments.

Dear Authors,

Congratulations on your manuscript. The reviewers were generally favorable of your approach and interested in your application. However, there were consistent inquiries about comparing your work to other simpler methods (such as k-means clustering) and increasing the clarity around your process for identifying the final clusters. We've included two reviews and appreciate your patience with the timing of your review. Best wishes,

We cannot make any decision about publication until we have seen the revised manuscript and your response to the reviewers' comments. Your revised manuscript is also likely to be sent to reviewers for further evaluation.

Sincerely,

Jennifer Wilson

Guest Editor

PLOS Computational Biology

Nina Fefferman

Deputy Editor

PLOS Computational Biology

Dear Authors,

Congratulations on your manuscript. The reviewers were generally favorable of your approach and interested in your application. However, there were consistent inquiries about comparing your work to other simpler methods (such as k-means clustering) and increasing the clarity around your process for identifying the final clusters. We've included two reviews and appreciate your patience with the timing of your review. Best wishes,

Reviewer's Responses to Questions

**Comments to the Authors:**

Reviewer #1: In this work, the authors report on a study that combines machine learning with comparative effectiveness analysis aimed at identifying which b/tsDMARDs would work “better” or “less” for patients subgroups. Two separate outcomes were used (1) non-response and (2) treatment response within a window of 15 months. For (1) it is a binary outcome defined as discontinuation of b/tsDMARD after physician’s recording of non-response. For (2) it is also a binary outcome defined as reduction in DAS28-esr by >=20%.  Additionally, the dataset included 3 classes of treatments (1) TNF inhibitor, (2) non-TNF inhibitor and (3) tsDMARD where each of the classes had a set of drugs associated with them. The reference group for the class effect analysis was the TNF inhibitor.

The authors used AnyNets (neural network architecture  http://ceur-ws.org/Vol-2926/paper2.pdf), specifically the autoencoder version to learn a latent representation from the encodings of the “present” features. These latent representations are used in a deep embedded clustering routine that involves a composite loss function. Clustering was done for 3, 4 and 5 predefined clusters. Additionally, they added “the time since measurement” to the features and repeated the 3, 4, 5 clustering to obtain a total of 24 clusters. Given that clusters were having similar features/characteristics, the authors regrouped them into 5 patient groups (based on the authors’ heuristics) and then extracted the most important features defining the groups for conducting a stratified analysis.

The first analysis used the 24 clusters to compare effectiveness of b/tsDMARDs using Cox proportional hazard regression to estimate hazard ratios of treatment no-response and treatment response separately. The results were compared among clusters belonging to same group (i.e. the groups they established above). Significant results present in the maximum number of clusters of the same group were reported. The same analysis was further repeated but this time using stratified groups based on the important features they determined earlier.

Overall, the study is interesting and informative and reports on “ad-hoc” approach that combines ML with regression analysis to establish treatment effect estimation.

The authors should address the following comments:

1- In many places in the writing, the authors would refer to “regrouping” or “selecting most distinctive features” etc. without providing details on the heuristics for selection. For example, what was the bases for regrouping in 5 groups, and how were the “few important” features selected for the validation analysis? It is worth describing the workflow; did the authors use comparisons/statistical tests for these decisions.

2- The use of AnyNet to cluster patients is an interesting choice. The authors should report on a baseline vanilla “k-means” or “hierarchical clustering” and compare the obtained groups. How many of the generated groups and subsequent analysis (of significant effects) still hold?

3- Did the authors consider the use and fitting of an “exposure model” that is useful for propensity score matching. In others words, the stratification would be based on the propensity scores generated from the exposure model.

4- The study does not deal with the dynamic (i.e. temporal) aspect of patients profiles during the 15-month window. Did the authors plan to incorporate dynamic information? This is something to discuss even if it is not planned for this study.

5- There are multiple studies that use ML for treatment effect estimation (i.e. individualised treatment effects). A pointer to the literature that is heavily researched in the last couple of years would further contextualise this work.

6- Minor comments : consider adding a visual/plot on how many times the reported variables were significant across all clusters (within groups)

Reviewer #2: I read with interest this paper on the identification of patient groups in rheumatoid arthritis by deep learning. This paper is a follow-up to a previous paper published in PLOS One in 2021 and referenced as reference 7 in this paper. The results only are of interest for 2 specific patient subgroups and rest on quite the usual criteria for response to treatment; however the approach warrants publication. Here are some comments to improve the paper.

1. Abstract results. The formulation of the observations at the end of the sentences makes it a little difficult to understand the results. Is it possible to reformulate?

2. Author summary page 3. I think it would be more useful to describe the five are a patient groups or at least to cite the elements defining these patient groups which are gender previous treatment and disease burden.

3. Methods, data source p5. The fact that the data refers to a very long period, 1999 to 2018, may have influenced the results since patient and physician expectations as well as the treatment armamentarium has changed a lot over this period. Could the authors please comment?

4. Exposures p6. It is surprising to me that the non TNFis are grouped in the methods, whereas they are reported separately in the results where the main results actually concern only Toci and Goli, representing only a small portion of the patients. Please comment.

5. Methods page 8 statistics. When describing the patient demographics and lifestyle factors on the top of page 8, please give the detail of which lifestyle factors were explored here.

6. Data analysis p9. I personally found the second paragraph on page 9 starting with stratification very complex to understand. Is it possible to simplify it?

7. Statistics page 10 first paragraph. It seems there was no external validation patient group. Is this correct? My understanding of artificial intelligence is that it is of key importance to validate the results on an external data set. Please consider this.

8. Page 10 results. The authors obtained 24 clusters. I believe there must have been a lot of overlap between the clusters. Is this the case? Please comment.

9. Five patient groups described page 10. It seems that the patient groups are defined mainly on gender and previous drug use. This is not very original. It is unfortunate that not more data were entered into the clusters. Could the authors please comment?

10. Drug use top of page 11. Could the authors comment if the use of the two conventional drugs was concomitant to the biologic, or was previous use. This is unclear from the text.

11. Smoking page 11. When was the smoking data collected ? Please clarify.

12. Response criteria. Using a change of DAS is not stringent. Why was this outcome chosen?

13. Disease activity and disease burden defined bottom of page 11 as predicted as outcomes. There is some circularity here since of course patients who do not do well will tend to not do well. Please comment.

14. Results page 12 on the 2 drugs of interest, Toci and Goli. How many patients were in fact in these drug clusters, I found this information in the table but please add it to the text.

15. Discussion page 15. The authors briefly commented on the inclusion of comorbidities – what woul that change?

16. Tables and figures. I suggest to add a figure with the percentage of response to adalimumab versus competitors for the main comparisons since currently it is very difficult to get an absolute you a response rather than a relative view of response.

17. Tables are very busy and difficult to read is there a way to simplify them? Perhaps be regrouping main significant results?

18. Minor comment. The word leflunomide takes an E and must be corrected throughout the paper.

**Have the authors made all data and (if applicable) computational code underlying the findings in their manuscript fully available?**

Reviewer #1: **No: **Sensitive data

Reviewer #2: **No: **

PLOS authors have the option to publish the peer review history of their article (what does this mean?). If published, this will include your full peer review and any attached files.

Reviewer #1: No

Reviewer #2: No
---

## [Editor Report · Decision Letter 1]

16 Dec 2022

Dear Dr Burkard

Thank you for your recent request to resubmit your manuscript, 'Patient Groups in Rheumatoid Arthritis Identified by Deep Learning Respond Differently to Biologic or Targeted Synthetic DMARDs', after accidental withdrawal. The Editors are willing to continue considering your paper and we are opening it up for you to make any necessary changes and resubmit. 

Please do not hesitate to contact us with any questions.

Best wishes,

Sarah Mayo

Sarah Mayo | Publications Assistant, PLOS Computational Biology

ploscompbiol@plos.org | Phone +44 (0) 1223-442824 

plos.org | ploscompbiol.org | @PLOSCompBiol

---

## [Decision Letter · Decision Letter 2]

4 Apr 2023

Dear Dr Burkard,

We are pleased to inform you that your manuscript 'Patient Groups in Rheumatoid Arthritis Identified by Deep Learning Respond Differently to Biologic or Targeted Synthetic DMARDs' has been provisionally accepted for publication in PLOS Computational Biology.

Reviewer #2 made some comments about relevant literature that I recommend addressing in the discussion section of the paper. But these were minor suggestions such that it was still suitable to accept your paper for publication.

Please ensure compliance with the PLOS Computational Biology code-sharing policy (https://journals.plos.org/ploscompbiol/s/code-availability) prior to publication. The policy states ways to be compliant while still commercializing the code.

Best regards,

Jennifer Wilson

Guest Editor

PLOS Computational Biology

Nina Fefferman

Section Editor

PLOS Computational Biology

Reviewer's Responses to Questions

**Comments to the Authors:**

Reviewer #1: Overall, the authors responded adequately to the comments raised. However, I come back to the 5th comment regarding the literature on treatment effect estimation with machine learning. There has been a great number of publications regarding treatment effect estimation (and causal learning) using clinical/medical data (including time series data) that might benefit the analysis in this current setup (such as considering time confounding variables and the temporal aspect of the data rather than having a fixed window type of analysis).

For example:

Van der Schaar lab https://www.vanderschaar-lab.com/

Feuerriegel Lab https://www.ai.bwl.uni-muenchen.de/publications/index.html

Sontag Lab http://clinicalml.org/

Reviewer #2: My comments have been dealt with

**Have the authors made all data and (if applicable) computational code underlying the findings in their manuscript fully available?**

Reviewer #1: **No: **

Reviewer #2: None

PLOS authors have the option to publish the peer review history of their article (what does this mean?). If published, this will include your full peer review and any attached files.

Reviewer #1: No

Reviewer #2: No

---

## [Editor Report · Acceptance letter]

11 May 2023

PCOMPBIOL-D-21-01954R2 

Patient Groups in Rheumatoid Arthritis Identified by Deep Learning Respond Differently to Biologic or Targeted Synthetic DMARDs

Dear Dr Burkard,

I am pleased to inform you that your manuscript has been formally accepted for publication in PLOS Computational Biology. Your manuscript is now with our production department and you will be notified of the publication date in due course.

With kind regards,

Zsofi Zombor
